# Influence of Antibacterial Coating and Mechanical and Chemical Treatment on the Surface Properties of PA12 Parts Manufactured with SLS and MJF Techniques in the Context of Medical Applications

**DOI:** 10.3390/ma16062405

**Published:** 2023-03-17

**Authors:** Anna Bazan, Paweł Turek, Andrzej Zakręcki

**Affiliations:** 1Faculty of Mechanical Engineering and Areonautics, Rzeszów University of Technology, Powstańców Warszawy 12, 35-959 Rzeszów, Poland; 2MEDIPRINTIC Sp. Z.O.O., Wojska Polskiego 9, 39-300 Mielec, Poland

**Keywords:** additive manufacturing, post-processing, antibacterial coating, surface topography, selective laser sintering (SLS), multi jet fusion (MJF), polyamide 12 (PA12)

## Abstract

Additive manufacturing (AM) is a rapidly growing branch of manufacturing techniques used, among others, in the medical industry. New machines and materials and additional processing methods are improved or developed. Due to the dynamic development of post-processing and its relative novelty, it has not yet been widely described in the literature. This study focuses on the surface topography (parameters *Sa*, *Sz*, *Sdq*, *Sds*, *Str*, *Sdr*) of biocompatible polyamide 12 (PA12) samples made by selective laser sintering (SLS) and multi jet fusion (MJF). The surfaces of the samples were modified by commercial methods: four types of smoothing treatments (two mechanical and two chemical), and two antibacterial coatings. The smoothing treatment decreased the values of all analyzed topography parameters. On average, the *Sa* of the SLS samples was 33% higher than that of the MJF samples. After mechanical treatment, *Sa* decreased by 42% and after chemical treatment by 80%. The reduction in *Sdq* and *Sdr* is reflected in a higher surface gloss. One antibacterial coating did not significantly modify the surface topography. The other coating had a smoothing effect on the surface. The results of the study can help in the development of manufacturing methodologies for parts made of PA12, e.g., in the medical industry.

## 1. Introduction

In the age of rapid development of the industry, additive techniques are used more often in manufacturing physical parts [1,2,3]. The principle of manufacturing the element is to add material layer by layer, thus forming the desired shape. Currently, many methods allow the production of physical objects in an additive manner [1,2]. The main differences between these methods result from the accuracy offered, the material used, the technique of applying subsequent layers, the speed of manufacturing elements, and the post-processing treatment used [2,4]. Currently, 3D printing techniques are used in many fields, including in the aviation industry [5,6,7], automotive industry [8,9], and in medicine [10,11,12], e.g., in the process of reconstructing and manufacturing the geometry of anatomical structures [13,14], surgical templates [15,16] and implants [17,18]. They are also often used in manufacturing orthoses [19,20,21] and prostheses [22,23].

Various materials are used in 3D printing [24,25,26]. Thermoplastics are used more often, as they are best suited for manufacturing final products and testing prototypes [27,28,29]. They have good mechanical properties and high resistance to impact, abrasion, and chemical resistance [27,30,31]. They can also be filled with carbon and glass, which improves their physical properties [30,31]. The most popular thermoplastics used in industry are polyamide (PA) [27,32] and thermoplastic polyurethane (TPU) [33,34], which often have better mechanical and physical properties than materials such as acrylonitrile butadiene styrene (ABS), polyethylene terephthalate glycol (PETG), or polylactic acid (PLA). In the case of thermosetting plastics (resins), they are better suited for applications where the aesthetics of the parts is essential [35,36]. They generally have high stiffness but are more brittle than thermoplastics, so they are not suitable for functional applications [37,38], except for resins that imitate the properties of ABS and polypropylene (PP) materials, which are intended for the production of dental inserts and implants [16,17].

Polyamide is used more often in parts production for the automotive [39] and aviation industries [40]. Polyamide is usually found in PA11 and PA12 variants. Polyamide is a solid and durable material [32,33]. It has a high melting point with a very low coefficient of friction, which is why it works great as a material for functional printing gears. Another critical property of polyamide is its hygroscopicity [41,42]. This aspect, in particular, facilitates the treatment of the surface with fabric dyes or spray paints to change the final aesthetics of the product. An essential feature of the material is its biocompatibility and the possibility of sterilization. Material from HP has additional Food and Drug Administration (FDA) certifications for skin contact, non-flammability, and toxicity. These properties make polyamide more often used in medicine [43] and orthoses production [44]. The most common technologies used in the process of 3D printing parts from polyamide are fused deposition modeling (FDM) [45], selective laser sintering (SLS) [46], and multi jet fusion (MJF) [47].

A crucial aspect in assessing the functionality of parts is the condition of the technological surface texture [48,49]. By selecting the appropriate technical parameters during the production of the element, the geometry of its surface is changed concerning the state under the technical conditions, ensuring the most favorable operating conditions, which are related to, among others, tightness, accuracy, a connection between elements (fit), wear or deformation. The created surface affects the wear of the cooperating parts and the thermal processes during operation. Examinations of the surface texture are currently being conducted to assess shape errors and surface roughness [50,51,52,53,54].

In the case of additive manufacturing of parts using polymeric materials, further post-processing of the surface is often necessary [55,56,57]. It consists inter alia on mechanical or chemical treatment [58]. The most commonly used methods include cleaning the surface, improving adhesion, and changing the surface in terms of utility and decoration. Polymeric materials’ surface modification methods improve the adhesion conditions of coatings, paints, and adhesives. In addition, tribological properties and physical properties (surface, corrosion, ultraviolet (UV) resistance) are modified. The applied additional post-processing treatment makes it possible to obtain the expected properties related to the condition of the surface layer [58,59]. Research on the state of the surface layer is fundamental not only for the aviation [48] or automotive industry [48] but more and more products made of polymeric materials are also used in medicine. They are not only used to produce illustrative models of anatomical structures [60] but surgical templates [61,62,63] are also made from polymer materials, which can have direct contact with human tissues and ready-made implants [63,64,65]. They are also used in the production of orthoses [66]. In the case of the production of orthoses, material extrusion technology is currently dominant. The most commonly used materials are ABS [44], polyethylene (PE) [67], PLA [68], PETG [69], or composites [70]. Due to the good functional properties of PA, this material is also increasingly used for the production of orthoses [44].

To obtain the appropriate properties of the surface roughness, tests are currently being carried out for PA12 material concerning the 2D [71,72,73] and 3D [74,75] parameters. However, there is no description of how the surface roughness changes as a result of different methods of post-processing treatment presented with quantified parameters. The proposed research is fundamental because currently, there are no developed standards for the surface treatment of parts 3D printed from polymeric materials used for the medical industry. The only guidelines that should be followed in the context of product approval for medical use are the following standards: ISO 13485:2016 [76] and 10993-1:2021 [77].

The paper presents test results estimating the surface topography of samples manufactured of polyamide PA12 material. The samples were made using the SLS and MJF methods. The surfaces of the samples were then subjected to mechanical and chemical treatment, and antibacterial coatings were applied. In addition, some samples were dyed. In the next step, the measurement process was carried out using a focus variation microscope. The obtained measurement data representing surface topographies were used to assess, among others, height, frequency, and hybrid parameters of the surface topography. The presented result is part of a broader research aimed at the development of orthosis manufacturing technology using additive technologies.

## 2. Materials and Methods

Rectangular samples with dimensions of 40 mm × 40 mm × 4 mm were subjected to surface roughness examinations. They were manufactured by SLS and MJF methods from polyamide PA12 and then subjected to post-processing (Figure 1). In the case of the SLS and MJF methods, standard 3D print settings for polyamide PA12 were used (Table 1). The samples were printed horizontally—the 40 mm × 40 mm surface was parallel to the XY plane.

The process of 3D printing test samples using the SLS method was carried out on the EOS P 396 printer. It consisted of the laser sintering particles of powdered thermoplastic polymer, then combining them into subsequent layers to form the final sample.

The HP MJF 5200 3D printer was used in the multi jet fusion manufacturing process. It deposits a layer of material in powder on the substrate. The ink head moves over the powder and applies a fixing and detailing agent. Then the infrared heating device moves across the print. The process is repeated until the entire model is formed layer by layer. Some of the samples thus manufactured by SLS and MJF were subjected further to post-processing. Samples without additional processing were referred to as reference samples (Ref).

To clean the surface of the test samples, mechanical and chemical treatments were used. For mechanical processing, a DyeMansion Powershot C was used (DM_PSC). It is equipped with a stainless steel rotating basket. Two simultaneously working sandblasting nozzles are placed perpendicularly to the rotating basket and its elements. Thanks to them, unbound material on the surface of the samples was removed during the process. Glass Beads (200–300 μm) were used in the process, pressure 3 bar for 5 min. As a result, a matte–gloss surface of the test samples was obtained.

For chemical surface treatment, a DyeMansion Powerfuse S machine was used. During the process, the entire process chamber was filled with steam under a vacuum. The treatment medium was VaporFuse Eco Fluid solvent. It was used to dissolve particles on the surface of the samples. As a result of the solvent used, the sample’s surface was smoothed. The solvent used was continuously circulated in a closed loop and was automatically recovered by the system. The solvent is currently approved for processing food contact plastics under Regulation (EU) 10/2011 [78]. The entire machining process took about 50 min. As a result of the applied treatment, the samples gained additional scratch resistance and water resistance.

Then, the process of dyeing the samples obtained after mechanical and chemical treatment was carried out on the DyeMansion DM60 coloring system. During a unique approach, which DyeMansion guards, the dye reacted with the material at high temperatures. In addition, under the influence of pressure, it penetrated the open pores of the test samples. This procedure allowed for uniform and permanent dyeing of the samples. The dyeing process took about 30 min and was carried out at a temperature of 115 ${}^{\circ }$C. As a result, colored test samples were obtained after mechanical (DM_PSC_Col) and chemical treatment (DM_VFS_Col).

In addition to DyeMansion, a PostPro SF50 from Additive Manufacturing Technologies (AMT) was also used in the chemical treatment process (samples AMT_CVS). This device works with the patented boundary layer automated smoothing technology (BLAST). It uses a method of chemical steam smoothing. Patented chemicals were used to smooth the printed surface of the samples. A suitably concentrated chemical solution sealed the surface to reduce its porosity. This treatment also prevented water penetration into the surface structure, thus improving the part’s mechanical properties.

In the case of applying antibacterial coatings, two solutions were used. In the former case, an antibacterial concentrate with the addition of silver (AC_Ag), Tarnamid MB AMB [79], was used, the base material of which is polyamide 6 produced by Grupa Azoty. The result of the research carried out by Grupa Azoty is a unique formula that allows for obtaining optimal antibacterial properties without compromising the processing and durability parameters of the materials. DAGlass developed the second solution (AC_Glass). The NANO-BARREN™ coating was created due to long-term research and experience gathered by the DAGLASS team [80]. Its specifics result from the use of magnetron technology. The nanolayers obtained by this method have a unique composition that ensures the reduction of colonies of bacteria and fungi without the need for additional UV irradiation. In both solutions, the manufacturers did not provide the parameters defining the process of applying antibacterial coatings.

The surface topography studies of the samples were carried out using the InfiniteFocus G4 focus variation microscope. The microscope is dedicated to measuring surfaces for which the *Ra* (*Sa*) of local roughness is above 10–15 nm. Five areas were measured on the top surface (near the corners and the center) for each type of sample with the parameters presented in Table 2. For each calculated pixel, information about the actual color is also recorded. Thanks to this, in addition to 3D data on the surface topography, 2D images in authentic colors are also obtained due to the measurement on the microscope.

The SPIP 6.4.2 software was used to process the measurement data and determine the surface topography parameters. The measurement data processing consisted of global leveling. The surface topography parameters were chosen for the primary profile. From a set of several dozen parameters, parameters that can be classified into four groups were selected for analysis:Height parameters *Sa* (arithmetical mean height) and *Sz* (maximum height) are determined according to the ISO 25178-2 standard [81]. Parameters are the most commonly used for assessing the surface topography, giving an overview of the differences in the height of unevenness on a given surface. The parameter *Sa* is the arithmetical mean height of the surface. This parameter is more averaging than *Sz*. The *Sz* parameter is the maximum height of the surface, i.e., the difference between the highest peak and the most significant valley.The parameters affecting the cosmetic appearance are *Sdq* (root mean square gradient) and *Sds* (density of summits). *Sdq* is determined according to ISO 25178-2 [81], and *Sds* according to ASME B46.1 [82]. The root mean square gradient is a general measurement of the slopes. It is a hybrid parameter depending on texture amplitude and spacing. With a similar value, the surface seems smoother if the *Sdq* is larger (the unevenness is more widely distributed). *Sdq* may be related to the degree of surface wetting by various fluid parameters. The density of summit is the number of peaks calculated based on hills per unit of the area.*Str* (texture aspect ratio) is determined according to the ISO 25178-2 [81]. The parameter measures isotropy. Parameters tend to be 0 for periodic surfaces with a dominant lay. For isotropic surfaces, its value approaches 1.*Sdr* (developed interfacial area ratio) is determined according to the ISO 25178-2 [81]. The parameter is helpful for surface coatings and adhesion tests. It expresses the increment of the interfacial surface area relative to an ideal plane in the size of the measurement region. It is a hybrid parameter dependent on texture amplitude and spacing. A surface with a lower *Sa* and finer spaced texture may have a higher *Sdr* value than a higher *Sa* but broader spaced texture.

On the top surface of the samples, in addition to the surface topography, hardness was also measured using the ZwickRoell 3105 hardness tester. The Shore D hardness test was performed according to standard [83]. Ten measurements were taken on each sample.

Statistical analyses of the research results were carried out using the package Pingouin written in Python. A significance level of 0.05 was assumed for all statistical tests.

## 3. Results and Discussion

### 3.1. Visual Assesment

Figure 2, Figure 3, Figure 4, Figure 5, Figure 6, Figure 7, Figure 8 and Figure 9 show fragments of the measured surface topographies of the samples made with the SLS and HP MJF methods. Different color palettes are used in the presented 3D views to facilitate the interpretation of individual views. The illustrated 3D maps are made with the same scale along the vertical axis to facilitate the comparison of other surfaces. Next to the 3D views are corresponding surface images in real colors. The surfaces of the MJF and SLS samples after analogous post-processing treatments have a similar character. Individual powder grains are visible on surfaces that have not been modified in any way (Ref) (Figure 2 and Figure 6). In the case of the SLS method, the visible grains have a more rounded shape and appear smoother.

AC_Ag and AC_Glass samples have antibacterial coatings (Figure 3 and Figure 7). The AC_Ag coating on the sample gives a metallic color and modifies the texture of the surface compared to its state before the coating was applied. It can be seen on 3D maps where individual grains are less distinct, while groups of grains stand out. The coating applied to the AC_Glass samples did not introduce significant changes in the 2D views of the tested surfaces.

On the DM_PSC and DM_PSC_Col samples (Figure 4 and Figure 8), mechanical smoothing was applied, and in the case of the DM_PSC_Col sample, additional dyeing was used. The surfaces of the DM_PSC and DM_PSC_Col samples are similar. The surfaces mainly consist of relatively large (compared to Ref samples), slightly convex areas. In the 2D and 3D views, grain boundaries are still visible in places. There are no apparent differences between the DM_PSC and DM_PSC_Col samples, apart from the color.

The DM_VFS_Col and AMT_CVS samples were chemically smoothed (Figure 5 and Figure 9). The DM_VFS_Col samples were previously additionally dyed. Compared to all other samples, grains of bonded powder are no longer visible on these surfaces. The surfaces consist of gentle peaks and valleys with a linear dimension of 50 µm. Smaller components of random roughness are visible (Figure 10) (in the 3D views in Figure 5 and Figure 9, these components resemble measurement noise).

### 3.2. Surface Topography Parameters

Figure 11, Figure 12, Figure 13, Figure 14, Figure 15 and Figure 16 presents graphs illustrating the measurement results of selected surface topography parameters. They enable the analysis of the impact of the manufacturing method (MJF and SLS) and the type of post-processing treatment on selected 3D surface parameters. Based on all the collected data, the one-way ANOVA tests with repeated measures were used to check the statistical significance of the influence of the manufacturing method and post-processing treatment on the topography parameters. In each test topography parameter was a dependent variable. When examining the impact of the manufacturing method on a given parameter, the manufacturing method was treated as a between-subjects factor and the type of post-processing as a within factor. When examining the impact of the type of post-processing on a given parameter, the type of post-processing was treated as a between-subjects factor and the manufacturing method as a within factor. The probability values determined in these tests are presented in Table 3.

The statistically significant influence of the manufacturing method was confirmed only for the height parameters *Sa* and *Sz* (Table 3). On average, the SLS samples had 22% higher *Sa* and *Sz* values than the MJF samples. For reference samples, i.e., without additional treatment, the value of the *Sa* parameter was about 33% higher, and for *Sz*, by 27% higher on SLS samples. Table 4 presents the relative differences of the height parameters *Ra* and *Sa* determined in the tests comparing SLS and MJF samples manufactured of PA12.

In addition to the results presented in [75], higher values of the parameters mentioned above were obtained for the SLS method. This may be influenced by an additional medium-binding powder grain in the MJF method. However, the values obtained in different studies should not be compared. In the research of Petzold et al. [74], parameter *Sa* changed in the range of 12.4–23.8 μm (i.e., almost twice) depending on the experimental conditions. In Cai’s studies, significant differences in *Ra* values (up to 43%) were also observed depending on the measurement site.

The type of post-processing had a significant impact on all analyzed parameters (Table 3). Figure 2, Figure 3, Figure 4, Figure 5, Figure 6, Figure 7, Figure 8 and Figure 9 show a general trend that these differences are mainly due to the type of smoothing treatment used. The values of the studied parameters are the highest for samples without smoothing treatment (Ref, AC_Ag, AC_Glass). The average values of the above parameters are related to the surface after mechanical treatment (DM_PSC, DM_PSC_Col) and the lowest for chemically smoothed samples (DM_VFS_Col, AMT_CVS). The changes in the surface topography parameters of the mechanically and chemically smoothed samples comparing to the reference samples are summarized in Table 5.

Similar to [75], chemical treatment reduced the roughness of the surface expressed by the *Sa* and *Sz* parameters by approximately 80%. In the tests carried out, machining had a much more significant effect on the surface topography than shown in [75]. For example, in the conducted research, the height parameters *Sa* and *Sz* decreased by over 42% and 23%, while in [75] by 7% and 13%, respectively. A similar trend can be observed when analyzing the *Sdq* and *Sdr* parameters. Reducing the values of these parameters is reflected in the bigger reflectivity of the surface and gloss—surfaces becomes less matte in visual perception.

Regardless of the type of smoothing treatment, the *Sds* value decreased by about 50%. After smoothing treatment, the elevation can be recognized in the place of the original occurrence of a group of grains that have been flattened to a lesser extent (as a result of mechanical treatment) or a greater extent (as a result of chemical treatment). Reducing the isotropic of the surface expressed by the *Str* parameter as a result of smoothing treatment may have the following justification. Before treatment, the structure was more random because it was strongly influenced by relatively numerous grains. After smoothing, there are fewer profile elements, peaks and valleys. It can be seen in the topography maps and the *Sds* parameter values. On the other hand, the probability of obtaining a perfectly random surface with fewer peaks and valleys is lower.

Using pairwise t-tests, the differences resulting from using different antibacterial coatings were examined (differences of each pair Ref–AC_Ag, Ref–AC_Glass, AC_Ag–AC_Glass were concerned). In each test, the topography parameter was a dependent variable, the type of post-processing was a between-subjects factor and the manufacturing method a within factor.

It was shown that only AC_Ag samples differ statistically from reference samples (without coating) and samples with DAGlass coating due to two parameters: *Sdq* and *Sdr*. It can therefore be concluded that the DA_Glass coating does not significantly modify the surface topography. The silver-containing coating had a more significant impact on the topography change in the SLS samples than the MJF samples. On the SLS samples, individual grains were more visible. The original differences in the height of the unevenness and the steepness of their slopes were more significant. The coating application had a “smoothing” effect on the surface.

The effect of staining was statistically tested by comparing the DM_PSC–DM_PSC_Col samples. The tests showed no differences in the parameter values resulting from using dyeing. Differences between the chemically treated samples DM_VFS_Col and AMT_CVS were also investigated using statistical tests. It has been shown that statistically, the values of the analyzed topography parameters in the samples, as mentioned earlier, are equal.

### 3.3. Surface Hardness

Taking hardness as the dependent variable, one-way ANOVA tests with repeated measures were carried out (similarly to the tests presented in Section 3.2). The tests showed that both the method of printing and the type of post-processing had a statistically significant effect on hardness. In Figure 17, it can be seen that the influence of the manufacturing method is not large (the scale on the vertical axis deliberately does not include 0, so the differences in the test samples can be seen). The SLS samples had an average of 0.6% higher hardness than the MJF samples (in absolute values by 0.43 Shore D).

When analyzing the effect of post-treatment, the previous analysis of variance was followed up with a Tukey test comparing each pair of samples after different treatments. No statistical differences were shown between the Ref, AC_Glass, DM_PSC, DM_PSC_Col and AMT_CVS samples. The average hardness of these samples was 75.3 Shore D. The DM_VFS_Col samples had a lower hardness than the others (72.6 Shore D). Samples with the antibacterial coating containing silver AC_Ag, on the other hand, had the highest hardness (76 Shore D).

In summarizing the results presented, it is worth recalling that the studies were conducted using process parameters recommended by the manufacturers of the solutions in question. The tests showed a statistically significantly higher hardness of the SLS samples than of the MJF samples, but these differences were not found to be significant in the context of the application of these additive technologies for the production of orthoses. Visually, the SLS samples were very similar to the MJF samples. Visually, the differences were mainly seen in the color of the samples. The tested parameters of the SLS samples had higher values than those of the MJF samples, but these differences were mainly in the samples that had not undergone a smoothing treatment. A smoothing treatment would have been recommended for hygienic reasons, ease of cleaning, and less tendency to accumulate dirt and irritate the skin.

Chemical treatment smooths the surface to a greater extent than mechanical treatment. The surface structure after mechanical treatment still contains significant cavities where impurities can accumulate. The surface of mechanically treated samples is matte and appears more homogeneous and flatter than that of chemically treated samples. Chemically treated samples are more reflective and also visually (not just to the touch) appear smoother.

The DAGlass antibacterial coating did not affect the visual aspect, hardness nor the microgeometry of the surface. The silver-containing coating significantly changes the optical properties of the surface and, in the case of the SLS samples, also had the effect of reducing the topography parameters. In the tests carried out, dyeing did not affect the surface topography parameters. It can be assumed that the type of pigment will only affect the visual aspect.

## 4. Conclusions

Additive techniques are becoming increasingly important in the medical industry. This is due to the rapid development of manufacturing methods, equipment used, materials, and post-processing techniques. One application of AM in the medical industry is the manufacture of orthoses. Biocompatible polyamide 12 and powder sintering-based technologies, such as SLS and MJF, are most commonly used for this purpose. The surfaces of PA12 parts printed using SLS and MJF processes can be subjected to various modifications to alter their functional properties. In the study, the flat surfaces of the samples produced by SLS and MJF were modified by mechanical treatment, chemical treatment, and the application of antibacterial coatings. In addition, dyeing was also included in selected processes, thus changing the color of the samples. The conclusions of the study are as follows:The type of manufacturing method (SLS or MJF) had a statistically significant effect only on the value of the height parameters *Sa* and *Sz*. A higher value of these parameters, on average by 22%, was observed for the SLS samples.The values of all analyzed parameters (*Sa*, *Sz*, *Sdq*, *Sds*, *Str*, *Sdr*) were highest for the samples without smoothing treatment (the reference sample and those with anti-bacterial coatings), medium after mechanical smoothing, and lowest after chemical smoothing.The chemical treatment reduced the roughness expressed by the *Sa* and *Sz* parameters by approximately 80%. After mechanical treatment, the height parameters *Sa* and *Sz* decreased by more than 42% and 23%, respectively.As a result of the mechanical and chemical smoothing treatment, the value of *Sdq* decreased by approximately 55% and 94%, respectively, and the parameter *Sdr* by 78% and 99%. The reduction in the values of these parameters is in the increased surface gloss—surface visually became less matte.Irrespective of the type of smoothing treatment, the density of summits *Sds* decreased by approximately 50%.There was no statistically significant effect of the type of chemical treatment on the topography parameter values.There was no statistically significant effect of dyeing on the topography parameter values.The antibacterial coating DAGlass had no statistically significant effect on topography parameter values.The antimicrobial coating containing silver had a larger effect on the topography of the SLS samples than MJF. This effect was of a ‘smoothing’ nature—the value of the analyzed parameters decreased.SLS samples had a hardness higher than MJF samples by an average of 0.43 Shore D.The hardness of samples DM_VFS_Col and AC_Ag were statistically different from the other samples, for which the average hardness was 75.3 Shore D. Sample DM_VFS_Col had the lowest hardness (72.6 Shore D). The application of a silver-containing antibacterial coating increased the hardness to 76 Shore D.

The presented research is part of the work related to the development of orthosis manufacturing technology using additive technologies. The presented results improve knowledge of the influence of smoothing post-processing and the application of antibacterial coatings on the surface topography of printed elements. On their basis, it is possible to infer, among other things, such functional properties as the ease of cleaning the surface, the tendency to accumulate impurities, or the rate of drying.

Potential benefits arise from bactericidal and fungicidal properties of antibacterial coating, which are linked to, among other things, its ability to absorb and dissipate moisture. As a result, the unpleasant odor emitted during increased activity is neutralized. At the same time, this increases the comfort of the orthosis, which is also influenced by the smoothing treatment. Moisture wicking and lower roughness reduce skin irritation, which in turn protect the skin from various infections. This aspect is particularly important for people with skin hypersensitivity or a tendency to allergies and sensitization. Due to recent years marked by the pandemic, people are more aware of microbiological risks. Therefore, the use of antibacterial protective measures also has an important psychological aspect. In this context, the use of an antibacterial coating and the ability to invoke its hygienic certification is an added value.

## Figures and Tables

**Figure 1 materials-16-02405-f001:**
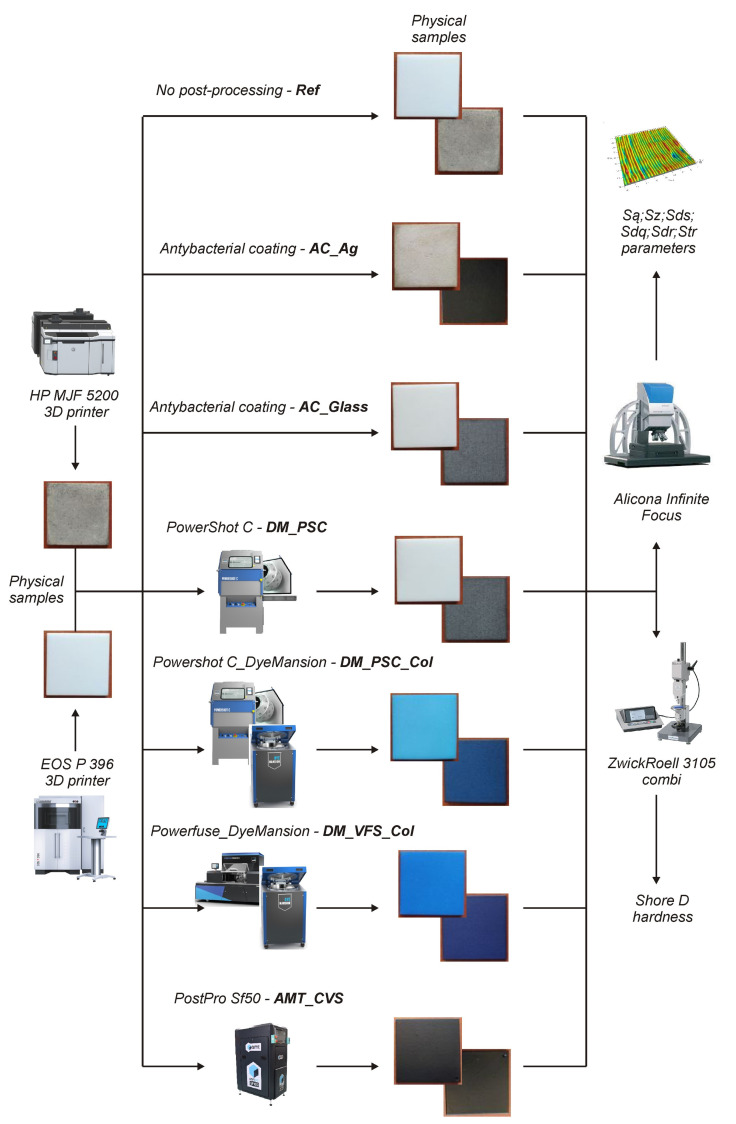
Scheme of research.

**Figure 2 materials-16-02405-f002:**
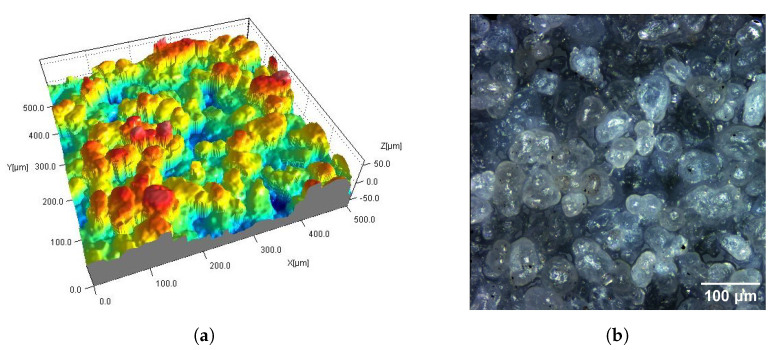
Surface of SLS sample without post-processing (Ref): fragment 0.5 mm × 0.5 mm of 3D map (**a**) and corresponding 2D view (**b**).

**Figure 3 materials-16-02405-f003:**
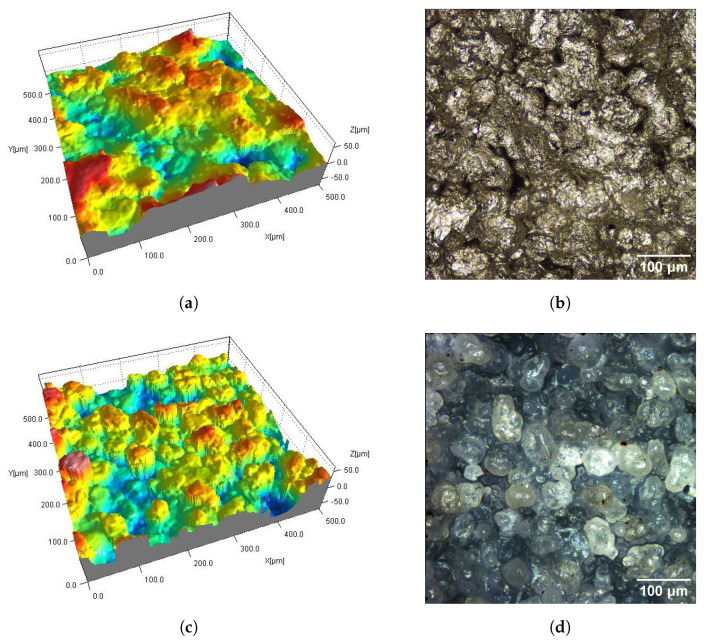
Surfaces of SLS samples with antibacterial coatings AC_Ag (**a**,**b**) and AC_Glass (**c**,**d**): fragments 0.5 mm × 0.5 mm of 3D maps (**a**,**c**) and corresponding 2D views (**b**,**d**).

**Figure 4 materials-16-02405-f004:**
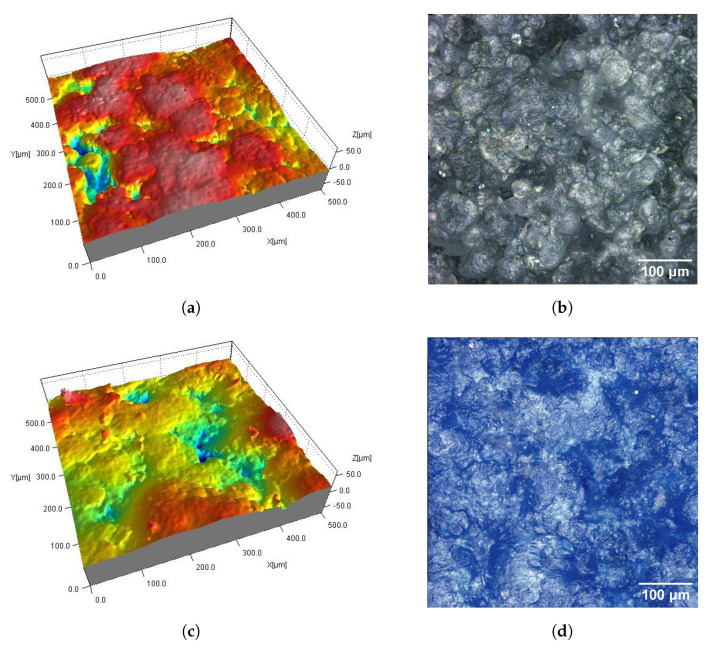
Surfaces of SLS samples after mechanical smoothing DM_PSC (**a**,**b**) and DM_PSC_Col (**c**,**d**): fragments 0.5 mm × 0.5 mm of 3D maps (**a**,**c**) and corresponding 2D views (**b**,**d**).

**Figure 5 materials-16-02405-f005:**
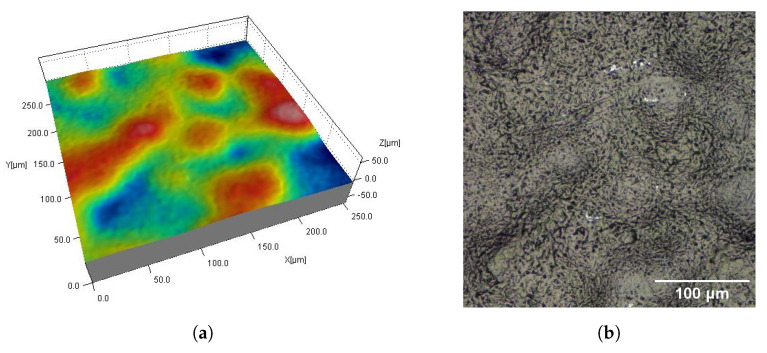
Surfaces of SLS samples after chemical smoothing DM_VFS_Col (**a**,**b**) and AMT_CVS (**c**,**d**): fragments 0.5 mm × 0.5 mm of 3D maps (**a**,**c**) and corresponding 2D views (**b**,**d**).

**Figure 6 materials-16-02405-f006:**
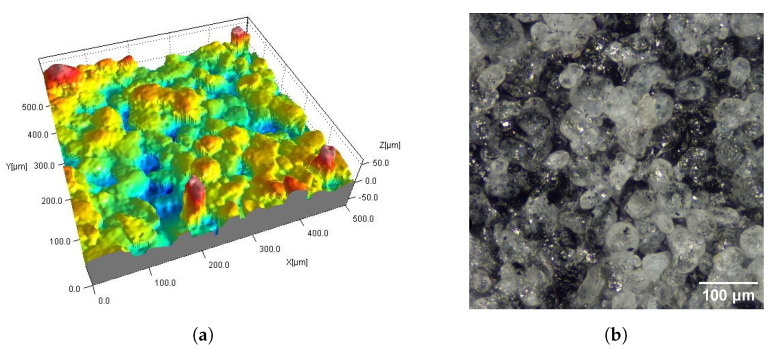
Surface of MJF sample without post-processing (Ref): fragment 0.5 mm × 0.5 mm of 3D map (**a**) and corresponding 2D view (**b**).

**Figure 7 materials-16-02405-f007:**
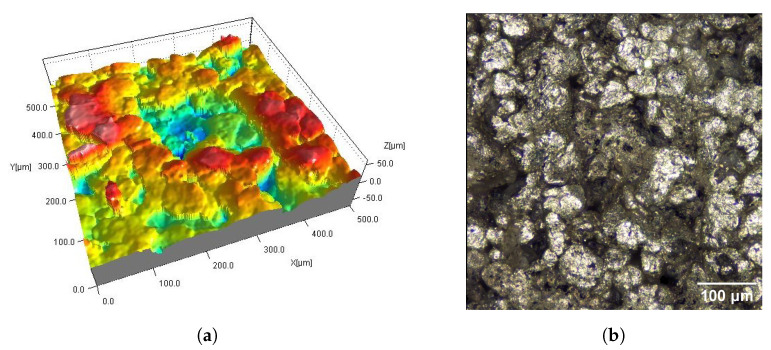
Surfaces of MJF samples with antibacterial coatings AC_Ag (**a**,**b**) and AC_Glass (**c**,**d**): fragments 0.5 mm × 0.5 mm of 3D maps (**a**,**c**) and corresponding 2D views (**b**,**d**).

**Figure 8 materials-16-02405-f008:**
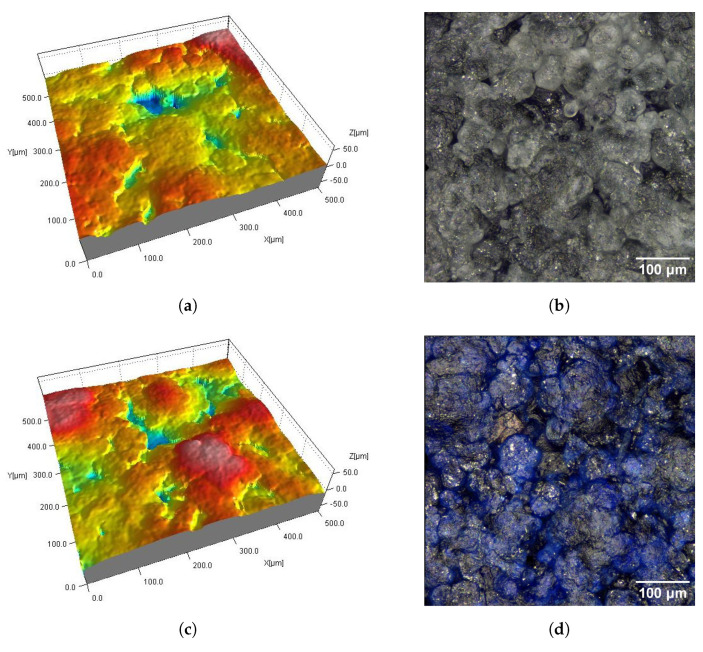
Surfaces of MJF samples after mechanical smoothing DM_PSC (**a**,**b**) and DM_PSC_Col (**c**,**d**): fragments 0.5 mm × 0.5 mm of 3D maps (**a**,**c**) and corresponding 2D views (**b**,**d**).

**Figure 9 materials-16-02405-f009:**
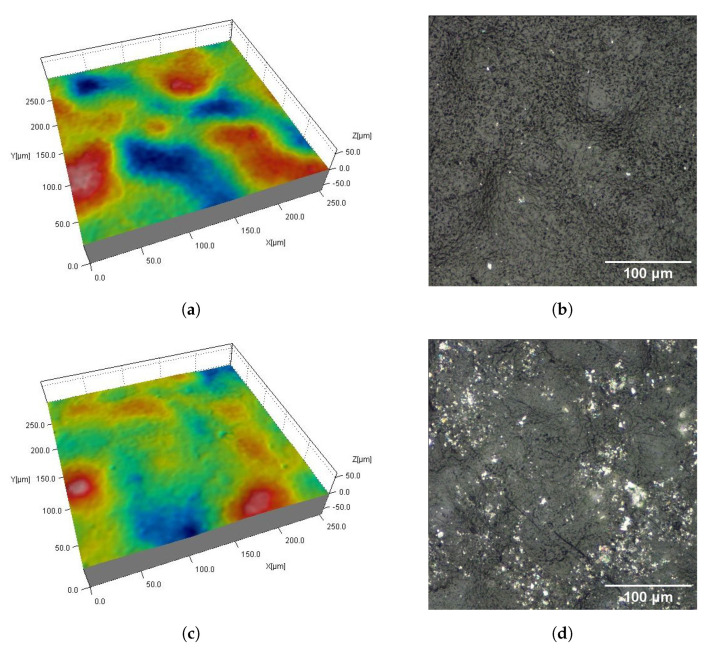
Surfaces of MJF samples after chemical smoothing DM_VFS_Col (**a**,**b**) and AMT_CVS (**c**,**d**): fragments 0.5 mm × 0.5 mm of 3D maps (**a**,**c**) and corresponding 2D views (**b**,**d**).

**Figure 10 materials-16-02405-f010:**
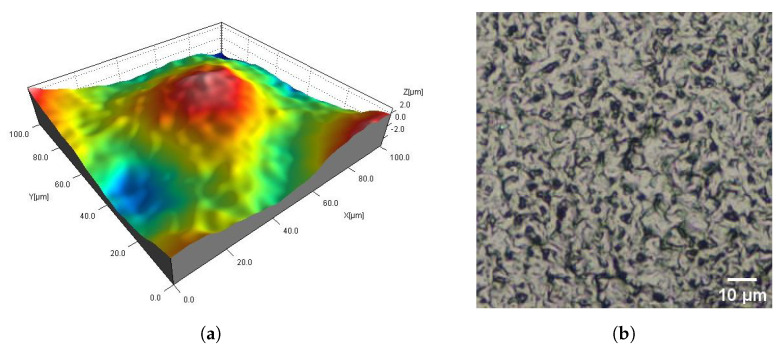
Fragment 0.1 mm × 0.1 mm of the SLS DM_VFS_Col sample surface: 3D map (**a**) and corresponding 2D view (**b**).

**Figure 11 materials-16-02405-f011:**
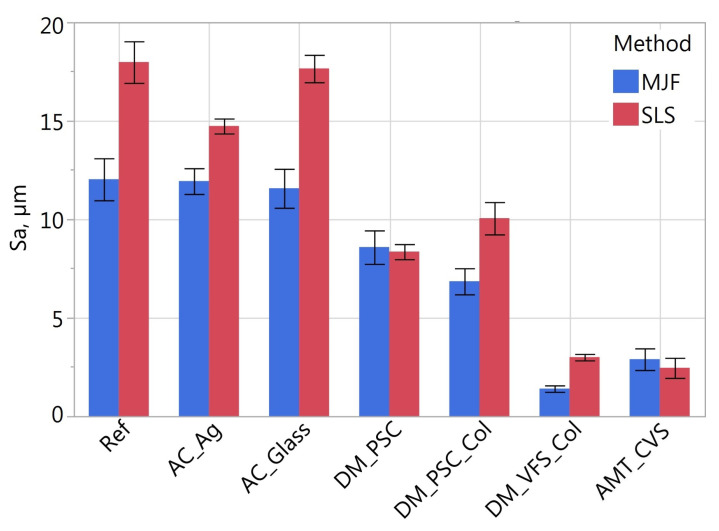
Arithmetical mean height *Sa* of measured samples.

**Figure 12 materials-16-02405-f012:**
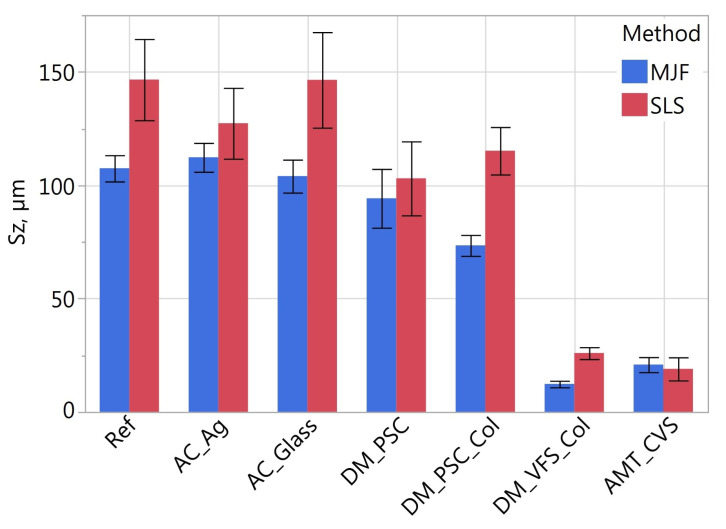
Maximum height *Sz* of measured samples.

**Figure 13 materials-16-02405-f013:**
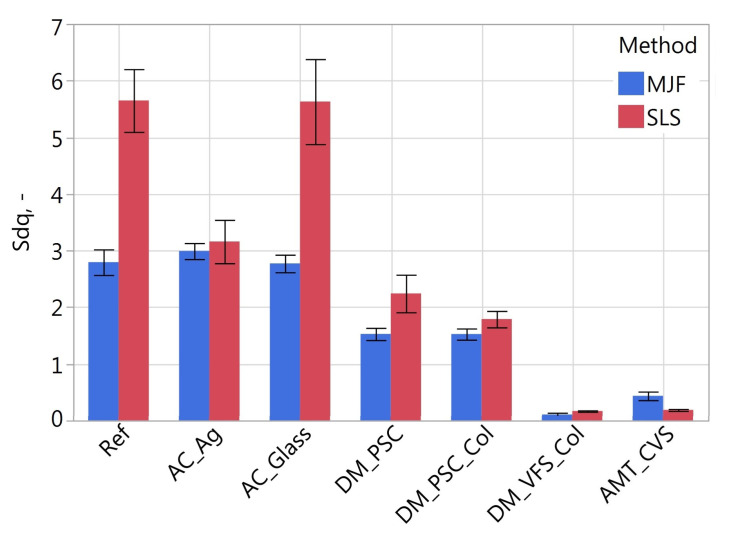
Root mean square gradient *Sdq* of measured samples.

**Figure 14 materials-16-02405-f014:**
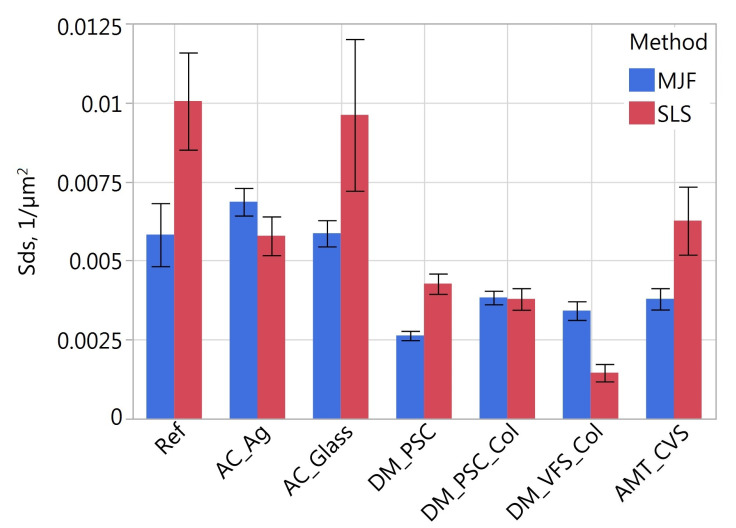
Density of summits *Sds* of measured samples.

**Figure 15 materials-16-02405-f015:**
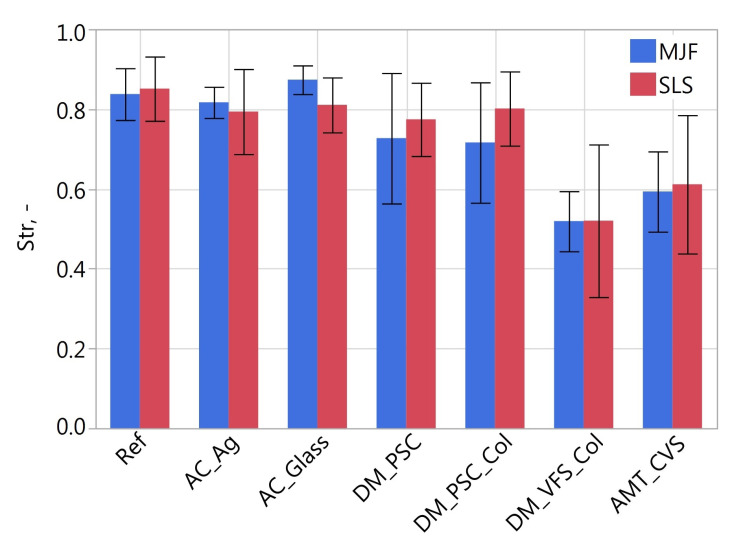
Texture aspect ratio *Str* of measured samples.

**Figure 16 materials-16-02405-f016:**
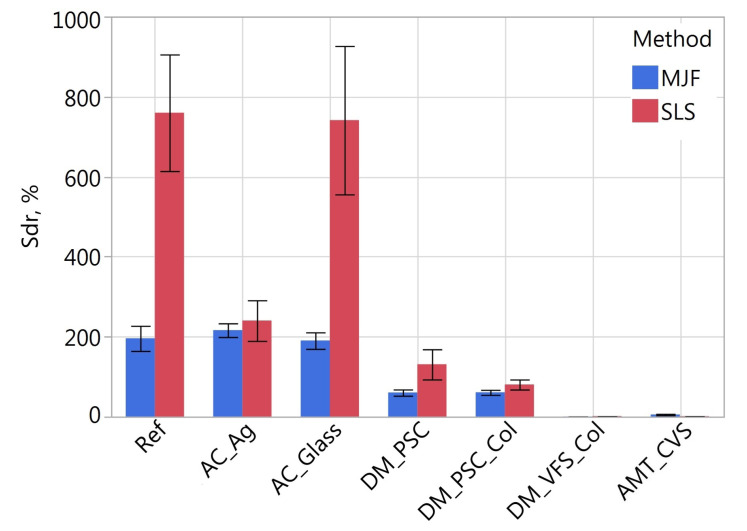
Developed interfacial area ratio *Sdr* of measured samples.

**Figure 17 materials-16-02405-f017:**
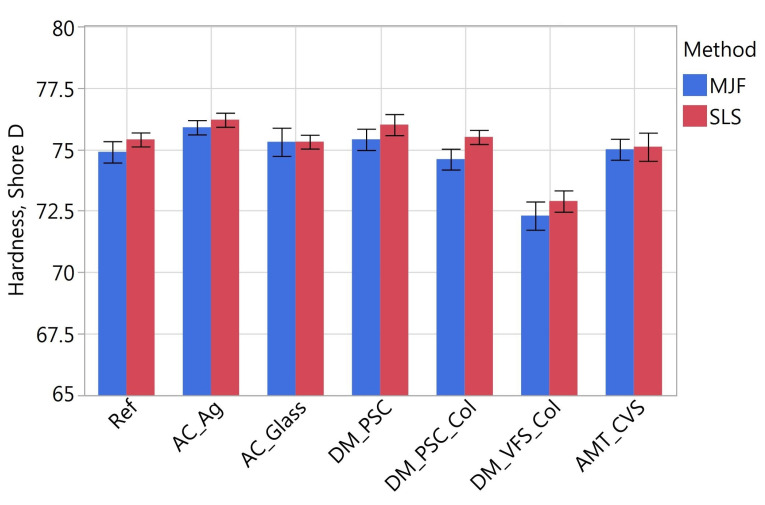
Hardness of MJF and SLS samples.

**Table 1 materials-16-02405-t001:** Parameters of manufacturing test samples.

MJF		SLS	
Printer	HP MJF 5200	Printer	EOS P 396
Building volume	380×284×380 mm${}^{3}$	Building volume	340×340×600 mm${}^{3}$
Building speed	Up to 5058 cm/ h	Scan speed	Up to 6 m/s
Layer thickness	0.08 mm	Layer thickness	0.12 mm
Print resolution (x, y)	1200 dpi	Laser type CO${}_{2}$	70 W

**Table 2 materials-16-02405-t002:** Measurement parameters of the surface topography on the focus variation microscope.

Samples	Ref, AC_Ag, AC_Glass, DM_PSC, DM_VFS_Col	DM_VFS_Col, AMT_CVS
Objective’s magnification	×20	×50
Number of image fields	2 × 2	4 × 5
Examination field	1 mm × 1 mm	1 mm × 1 mm
Vertical resolution	100 nm	25 nm
LaterAL resolution	2.93 μm	2.13 μm
Pixel size	0.44 μm × 0.44 μm	0.35 μm × 0.35 μm
Mean repeatability	34 nm	5 nm

**Table 3 materials-16-02405-t003:** Probability values (*p*-values) obtained as a result of repeated measures ANOVA examining the significance of the impact of the manufacturing method and type of post-processing treatment on surface topography parameters.

Between Factor	Within Factor	Dependent Variable					
		*Sa*	*Sz*	*Sds*	*Sdq*	*Sdr*	*Str*
manufacturing method	type of post-processing	0.015	0.006	0.115	0.065	0.081	0.668
type of post-processing	manufacturing method	0.0005	0.0002	0.04	0.01	0.001	0.001

**Table 4 materials-16-02405-t004:** Mean values of *Ra* and *Sa* parameters of samples made by SLS and MJF and the relative difference *RD* between them (comparing to SLS samples).

Source	Parameter	SLS	MJF	*RD*
This research	*Sa*	17.98 μm	12.03 μm	33.09%
DyeMansion [75]	*Sa*	8.45 μm	10.80 μm	−27.81%
DyeMansion [75]	*Ra*	7.67 μm	9.79 μm	−27.64%
Rosso et al. [84]	*Ra*	12.06 μm	11.06 μm	8.29%
Cai et al. [85]	*Ra*	25.66 μm	15.66 μm	38.97%

**Table 5 materials-16-02405-t005:** Relative difference *RD* in the values of surface topography parameters after smoothing comparing to reference samples.

Source	Smoothing Method	*RD*(Sa), %	*RD*(Sz), %	*RD*(Sdq), %	*RD*(Sds), %	*RD*(Sdr), %	*RD*(Str), %
This research	mechanical	−42.3	−23.8	−54.8	−52.2	−77.6	−10.7
This research	chemical	−83.5	−84.6	−93.6	−49.8	−99.2	−33.6
DyeMansion [75]	mechanical	−7.3	−13.1	−26.9	-	−45.8	-
DyeMansion [75]	chemical	−77.9	−83.1	−60.7	-	−84.1	-

## Data Availability

Data can be made available on request.

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
