# Peer review of "Influence of Antibacterial Coating and Mechanical and Chemical Treatment on the Surface Properties of PA12 Parts Manufactured with SLS and MJF Techniques in the Context of Medical Applications"

_materials, 2023, doi:10.3390/ma16062405_

Round 1

Reviewer 1 Report

firstly, the title of the manuscript  is not explicite. "various modifications of models" is not meaningful  at all; the used printing material should be precised specificily;

secondly, the surface morphology analysis zone should be precsied in considering the printing process. The corresponding analysis precision should be discussed in the part 2;

thirdly,  in figure 2, 3, 4, 5, 6, 7 8, 9 and 10, it is necessary to precise what correspond to each photo and figure with right caption;  each photo should be presented with an observation scale; 

Author Response

Dear Reviewer:

I would first like to thank you for the comments concerning our manuscript. The comments were all valuable and helpful for revising and improving our paper . We have tried our best to revise the article as recommended. I hope to be met with approval.

Thanks very much again for your attention to our manuscript. Below are the responses to the comments.

  1. The title of the manuscript  is not explicite. "various modifications of models" is not meaningful  at all; the used printing material should be precised specificily;

Response: The title of the paper was modified to be more precise and explicit: “Influence of antibacterial coating and mechanical and chemical treatment on surface properties of PA12 parts manufactured with SLS and MJF techniques in the context of medical applications”

  1. The surface morphology analysis zone should be precised in considering the printing process. The corresponding analysis precision should be discussed in the part 2;

Response: Additional information was provided in the text. The samples were printed horizontally - the 40 mm x 40 mm surface was parallel to the XY plane. All measurements were taken on the top surface – near the corners and in the center. The InfiniteFocus microscope is dedicated to measuring surfaces for which the Ra (Sa) of local roughness is above 10-15 nm. Mean repeatability when using objective x20 was equal to 34 nm, and 5 nm for x50 objective.

  1. In figure 2, 3, 4, 5, 6, 7 8, 9 and 10, it is necessary to precise what correspond to each photo and figure with right caption;  each photo should be presented with an observation scale; 

Response: Captions of figures 2-10 were modified. Observation scale was added.

Reviewer 2 Report

The additive manufacturing of medical parts has a large potential for application. As a result, the investigation of the microstructure and properties of the additively manufactured samples for bioapplication is an actual topic. In the paper "Analysis of the surface topography after various modifications of models manufactured with SLS and MJF techniques in the context of medical applications," the influence of printing type and post-printing surface treatment on the surface parameters was investigated. The authors have shown that chemical treatment has a more significant influence on roughness than mechanical treatment. The authors have made well experimental work and discussed the results at a high scientific level. However, in my opinion, some points of the paper should be improved accordingly following comments:

1.                  Some descriptions of the additive manufacturing techniques and methods of investigation in the paper are well-known excessive. It is recommended to remove/revise Lines 20-24, 100-111, etc.

2.                  The authors have used standard 3D print settings for polyamide PA12 for both producing techniques. It is recommended to compare the energy density achieved at such parameters in SLS and MJF methods. The energy density has a significant influence on the microstructure of the printed samples including the surface roughness. It is recommended to analyze the influence of the energy density level on the surface parameters of the as-printed samples.

3.                  The authors have investigated only the surface properties of the printed and treated samples. However, internal porosity has also may significantly influence the functionality of the medical parts. It is recommended to analyze the influence of the printing type on internal porosity.

4.                  The functionality of the surface depends not only on the roughness but also on the mechanical properties. It is recommended to estimate the microhardness of the investigated samples after different treatments.

5.                  The investigated area for all samples was only 0.25 μm2. It is too small in comparison with the size of the sample. It may impugn the reproducibility of the results. It is recommended to investigate 3-5 different zones for some samples to approve the statistical value of the obtained results.

6.                  Unfortunately, the absence of the antibacterial investigation of the obtained material limits the practical application of the authors’ results. It is recommended to conclude in the last part the potential advances in the antibacterial effect of the developed technology in comparison with usual materials.

Author Response

Dear Reviewer:

I would first like to thank you for the comments concerning our manuscript. The comments were all valuable and helpful for revising and improving our paper and the important guiding significance to our research. We have tried our best to revise the article as recommended. I hope to be met with approval.

Thanks very much again for your attention to our manuscript. Below are the responses to the comments.

  1. Some descriptions of the additive manufacturing techniques and methods of investigation in the paper are well-known excessive. It is recommended to remove/revise Lines 20-24, 100-111, etc.

Response: Lines 20-24 were crossed out. Lines 100-111 were revised.

  1. The authors have used standard 3D print settings for polyamide PA12 for both producing techniques. It is recommended to compare the energy density achieved at such parameters in SLS and MJF methods. The energy density has a significant influence on the microstructure of the printed samples including the surface roughness. It is recommended to analyze the influence of the energy density level on the surface parameters of the as-printed samples.

Response: Thank you very much for drawing attention to the parameter energy density in the context of comparing SLS and MJF methods. We have calculated the energy delivered to the surface in the case of the SLS samples. The energy density was 0.03(8) J/mm2 (0.32 J/mm3). However, we do not have sufficient data on the MJF process to be able to count the energy in this method.   We also did not find the necessary data in the literature. We also did not find any article in which the authors provided the energy value or related parameters (e.g. lap power) used in their study. Surprisingly, many of these articles were devoted to the mechanical properties of the components produced by the MJF method or to FEM analyses.

We are currently in the process of strength testing. This makes us all the more grateful for your valuable comment. We hope to be able to include a parameter related to the energy of the bonding process in future studies.

  1. The authors have investigated only the surface properties of the printed and treated samples. However, internal porosity has also may significantly influence the functionality of the medical parts. It is recommended to analyze the influence of the printing type on internal porosity.

Response: In the research presented here, we focused our attention on the additional surface treatment of the samples. Therefore, we analyzed the properties of the surface only - expressed by surface texture parameters. After taking note of Note 4, the research was extended to include hardness measurements.

We are currently in the process of strength and ageing tests. However, in developing an analysis of the research material collected to date, we have found it to be quite extensive. Of course, we agree, with the reviewer, that the analysis presented is not exhaustive, hence our plans for further research. Topically, the issue of internal porosity corresponds more closely to the next stage of our research.

  1. The functionality of the surface depends not only on the roughness but also on the mechanical properties. It is recommended to estimate the microhardness of the investigated samples after different treatments.

Response: Thank you very much for your valuable comment. We agree that, when analyzing the impact of surface treatment,  it is worth considering more than just geometrical parameters. The research was extended to include hardness measurement.

  1. The investigated area for all samples was only 0.25 μm2. It is too small in comparison with the size of the sample. It may impugn the reproducibility of the results. It is recommended to investigate 3-5 different zones for some samples to approve the statistical value of the obtained results.

Response: The investigation area was 1 mm x 1 mm (examination area mentioned in Table 2). Each sample was measured in 5 different zones.

  1. Unfortunately, the absence of the antibacterial investigation of the obtained material limits the practical application of the authors’ results. It is recommended to conclude in the last part the potential advances in the antibacterial effect of the developed technology in comparison with usual materials.

Response: The article is complemented by a description of the potential advantages of applying antibacterial coatings. Potential benefits arise from bactericidal and fungicidal properties of antibacterial coating, which are linked to, among other things, its ability to absorb and dissipate moisture. As a result, the unpleasant odor emitted during increased activity is neutralized. At the same time, this increases the comfort of the orthosis. Moisture wicking reduces skin irritation, which in turn protects the skin from various infections. This aspect is particularly important for people with skin hypersensitivity or a tendency to allergies and sensitization. Due to recent years marked by pandemics, people are more aware of microbiological risks. Therefore, the use of antibacterial protective measures also has an important psychological aspect. In this context, the use of an antibacterial coating and the ability to invoke its hygienic certification is an added value.

Reviewer 3 Report

“Analysis of the surface topography after various modifications of models manufactured with SLS and MJF techniques in the context of medical applications “investigates the effectiveness of different post- process treatments on samples manufactured with plyammide 12 via additive techniques. Surface parameters are used as quality indices of the performed mechanical and chemical treatments on the sample’s surfaces.

Revision or comment

The work is interesting and in a broader application field of the additive manufacturing and, as aim of the presented research, in the medical area. Different techniques and post processing experimentation were performed and presented with the corresponding analysis.

To better highlight the activity, authors should better describe the statistical analysis performed and try to make some section more fluent.  

At the end of the introduction there is no a brief presentation of the work that will be presented in the following sections.

Moreover, some improvements can be implemented.

Authors are invited to inspect the paper for minor revisions like that follow and to check for English language:

Line 19: the product of a manufacturing process is not a physical model! Please, choose a different term to indicate an object (sample, component, part…) every time that is used in the paper.

Line 52: Delete “treated”.

Line 76: the sentence “Research on the …” should not be divided from the consecutive “More and more…” by a full stop.

Line 111:  It is not immediately clear that samples with no post processing are referred as “Ref”

Line 137: Does The acronym “AMT” refer to two different items?

Line 225-226: Which test do you perform exactly? 2 sample t test? A DOE analysis? In this case what are the dependent variables?

Line 244: Figure xxxx?

Line 270: 2 sample t test?

Author Response

Dear Reviewer:

I would first like to thank you for the comments concerning our manuscript. The comments were all valuable and helpful for revising and improving our paper and the important guiding significance to our research. We have tried our best to revise the article as recommended. I hope to be met with approval.

Thanks very much again for your attention to our manuscript. Below are the responses to the comments.

  1. To better highlight the activity, authors should better describe the statistical analysis performed and try to make some section more fluent.  

Response: We corrected the names of the statistical tests and added explanations of how they were carried out. A section-by-section breakdown of the topography parameter analysis that confused has been removed.

  1. At the end of the introduction there is no a brief presentation of the work that will be presented in the following sections.

Response: The presentation of the work was added to the introduction.

  1. Line 19: the product of a manufacturing process is not a physical model! Please, choose a different term to indicate an object (sample, component, part…) every time that is used in the paper.

Response: All 'model' phrases that were inappropriately used have been replaced with other terms.

  1. Line 52: Delete “treated”.

Response: Indicated word was deleted.

  1. Line 76: the sentence “Research on the …” should not be divided from the consecutive “More and more…” by a full stop.

Response: Sentences were joined.

  1. Line 111:  It is not immediately clear that samples with no post processing are referred as “Ref”

Response: Indicated text was rewritten.

  1. Line 137: Does The acronym “AMT” refer to two different items?

Response: Yes, we made a mistake. We renamed samples after chemical treatment using AMT technology.

  1. Line 225-226: Which test do you perform exactly? 2 sample t test? A DOE analysis? In this case what are the dependent variables?

Response: Sorry for the confusion. Sens of the text was lost in translation. In this case several tests were carried out. Each of them was one-way ANOVA with repeated measurements, and a dependent variable was one of the surface topography parameters. The name of the test has been corrected in the text. And additional information about dependent variable was introduced.

  1. Line 244: Figure xxxx?

Response: Figures numbers were added.

  1. Line 270: 2 sample t test?

Response:  We have corrected the name of the test. In this case pairwise t-tests were used.

Round 2

Reviewer 1 Report

in the revised version of the manuscript, author has made necessary corrrections including nearly all remaarks from reviewers.

But the discussion part can be strengthened to leave a global synthesis about different results and explain the results evolution;

Author Response

Thank you very much for your comment. Indeed, the Results and Discussion section is quite extensive and it is worth synthesizing it. The changes made are indicated in the text in yellow.

Reviewer 2 Report

The authors have answered previous comments the paper may be accepted for publication.

Author Response

Again, thank you for your effort in analyzing our paper and all suggestions made to improve it.